# Reciprocal synapses between mushroom body and dopamine neurons form a positive feedback loop required for learning

**Isaac Cervantes-Sandoval\*, Anna Phan, Molee Chakraborty, Ronald L Davis\***

Department of Neuroscience, The Scripps Research Institute Florida, Jupiter, United States

**Abstract** Current thought envisions dopamine neurons conveying the reinforcing effect of the unconditioned stimulus during associative learning to the axons of *Drosophila* mushroom body Kenyon cells for normal olfactory learning. Here, we show using functional GFP reconstitution experiments that Kenyon cells and dopamine neurons from axoaxonic reciprocal synapses. The dopamine neurons receive cholinergic input via nicotinic acetylcholine receptors from the Kenyon cells; knocking down these receptors impairs olfactory learning revealing the importance of these receptors at the synapse. Blocking the synaptic output of Kenyon cells during olfactory conditioning reduces presynaptic calcium transients in dopamine neurons, a finding consistent with reciprocal communication. Moreover, silencing Kenyon cells decreases the normal chronic activity of the dopamine neurons. Our results reveal a new and critical role for positive feedback onto dopamine neurons through reciprocal connections with Kenyon cells for normal olfactory learning.

**\*For correspondence:** isandova@ scripps.edu (IC-S); rdavis@scripps. edu (RLD)

**Competing interests:** The authors declare that no competing interests exist.

## Introduction

Molecular, cellular and systems neuroscience studies using *Drosophila* have uncovered several major tenets that reveal the logic by which olfactory memories are formed and stored. The first is that modulation of mushroom body (MB) Kenyon cell (KC) synaptic activity underlies associative learning and memory (*Davis, 1993*, *Davis, 2011*; *Heisenberg, 2003*; *Busto et al., 2010*). KCs are connected and respond to multiple primary sensory centers including those for olfaction, vision and gustation (*Yagi et al., 2016*). The major olfactory input occurs in the MB calyx, with the KC dendrites receiving input in this neuropil region from approximately 180 olfactory projection neurons. The representation of different odorants is highly sparse (*Honegger et al., 2011*) across the three major types of KC – the $\alpha\beta$, $\alpha'\beta'$ and $\gamma$ – being generated, in part, by a negative feedback loop between KC and the GABAergic APL neuron (*Liu and Davis, 2009*; *Lin et al., 2014a*).

A second tenet is that the modulation of Kenyon cell function for olfactory learning employs intracellular cAMP signaling. Multiple learning mutants that alter cAMP synthesis, degradation, or effector function have been characterized and mapped through expression or functional studies to the Kenyon cells (*Davis, 1993*; *2005*). More specifically, inputs representing an odor-conditioned stimulus (CS) along with an unconditioned stimulus (US) to the KC lead to the mobilization of the cAMP signaling cascade (*Davis, 1993*) employing the Rutabaga-encoded adenylyl cyclase as a coincidence detector (*Tomchik and Davis, 2009*) to induce changes in the output of the KC.

A third tenet is that the mushroom body output neurons (MBOn) receive input from the KC and their activation influences approach or avoidance behavior (*Davis, 1993*; *Heisenberg, 2003*; *Aso et al., 2014b*; *Owald et al., 2015*). There are 21 classes of MBOn whose dendrites tile the lobes

containing the axons of the KC in 15 different compartments. Recent studies have shown that learning alters odor drive to specific MBOn (*Séjourné et al., 2011*; *Pai et al., 2013*; *Plaçais et al., 2013*; *Bouzaiane et al., 2015*; *Hige et al., 2015*; *Owald et al., 2015*). Interestingly, reward learning appears to reduce drive to output pathways that direct avoidance, whereas aversive learning increases drive to avoidance pathways while reducing drive to approach pathways.

A fourth tenet is that dopamine neurons (DAn) are activated by aversive or rewarding stimuli to provide the US input for olfactory classical conditioning (*Riemensperger et al., 2005*; *Mao and Davis, 2009*; *Liu et al., 2012*). There exist several lines of evidence in support of this model. First, blocking DAn during training with temperature sensitive dynamin (shibire) impairs memory acquisition (*Schwaerzel et al., 2003*). Second, mutants for the dopamine receptor dDA1 impair memory acquisition that can be rescued by expressing this receptor in KC (*Kim et al., 2007*; *Qin et al., 2012*). Third, the artificial activation of DAn using thermo- or optogenetic approaches along with odor CS stimulation leads to conditioned behavior (*Schroll et al., 2006*; *Claridge-Chang et al., 2009*; *Aso et al., 2010, 2012*; *Burke et al., 2012*; *Liu et al., 2012*). Thus, artificial stimulation of the DAn is sufficient to signal the reinforcing effect of the US. The innervation by DAn occurs primarily on the axonal processes of the KC in the MB lobes, across the same 15 compartments occupied by the dendrites of the MBOn (*Mao and Davis, 2009*; *Pech et al., 2013*; *Aso et al., 2014a*). These findings have suggested that DAn stimulation, driven by a US, changes the odor-specific output weight of KC synapses onto the corresponding MBOn, tilting the MBOn network to direct appropriate behavior. The dopaminergic inputs modulate KC synaptic transmission with precise spatial specificity, allowing KC to differentially convey olfactory signals to each of their postsynaptic MBOn (*Cohn et al., 2015*).

However, this model of multiple, independent plasticity/output channels is oversimplified given evidence of additional and poorly understood computations occurring within the KC neuropil. For instance, *Perisse et al. (2016)* provided evidence that plasticity in the MBOn-γ1pedc compartment initiates feed-forward inhibition that alters the flow of information in the MBOn$\beta'$2mp and MBOnγ5$\beta'$2a compartments. In addition, different DAn drive synaptic changes between KC and MBOn with different rules (*Hige et al., 2015*). Activation of γ1pedc DAn induces both learning and odor-specific synaptic depression in MBOnγ1pedc when paired with only a 1 s odor presentation, whereas similar plasticity in the α2α'2 compartment requires 1 min odor presentation with DAn activation.

Here, we report additional complexity in the circuitry and connections that influence olfactory learning. We demonstrate that DAn are both pre- and postsynaptic to KC, forming axoaxonic reciprocal synapses that are required for optimal learning through a positive feedback mechanism. DAn input to KC feeds back onto the same DAn through KC>DAn cholinergic synapses amplifying the synaptic release of DA required for reinforcement.

## Results and discussion

### KC are both pre-and post-synaptic to DAn forming axoaxonic reciprocal connections

We previously showed that genetic insults in KC affect ongoing activity in DA PPL1 neurons, the set of DAn that innervate the vertical lobes, heel, and junction area of the MB (*Figure 1A,B*; compartments γ1–2, α1–3, α'1–3). Reducing the expression of the Scribble protein using the KC-specific *R13F02-gal4* driver increases total ongoing activity measured in the axon terminals of the DAn that innervate γ2α'1 (*Cervantes-Sandoval et al., 2016*). This result suggested the existence of a retrograde signal from the postsynaptic KC to the presynaptic DAn partners that influences their activity, or that KC might be both pre- and post-synaptic to DAn, perhaps forming axoaxonic reciprocal synapses in the complex MB lobe neuropil. We first used trans-synaptic GFP fluorescence reconstitution (*Macpherson et al., 2015*) to look for a possible physical and functional connection between KC axons and DAn axons. This system uses the expression of a split version of GFP tethered to presynaptic synaptobrevin (*syb::spGFP1-10*) in the hypothetical presynaptic partner and a membrane tethered version of the complementing split GFP (*CD4::spGFP11*) in the hypothetical postsynaptic counterpart (*Figure 1C*). *Figure 1B* shows KC co-labeled with DAn represented by the *TH-LexA* driver innervating the α'3, α3, α2α'2, γ1ped, γ1, γ2α'1 and β2 and β'2 compartments

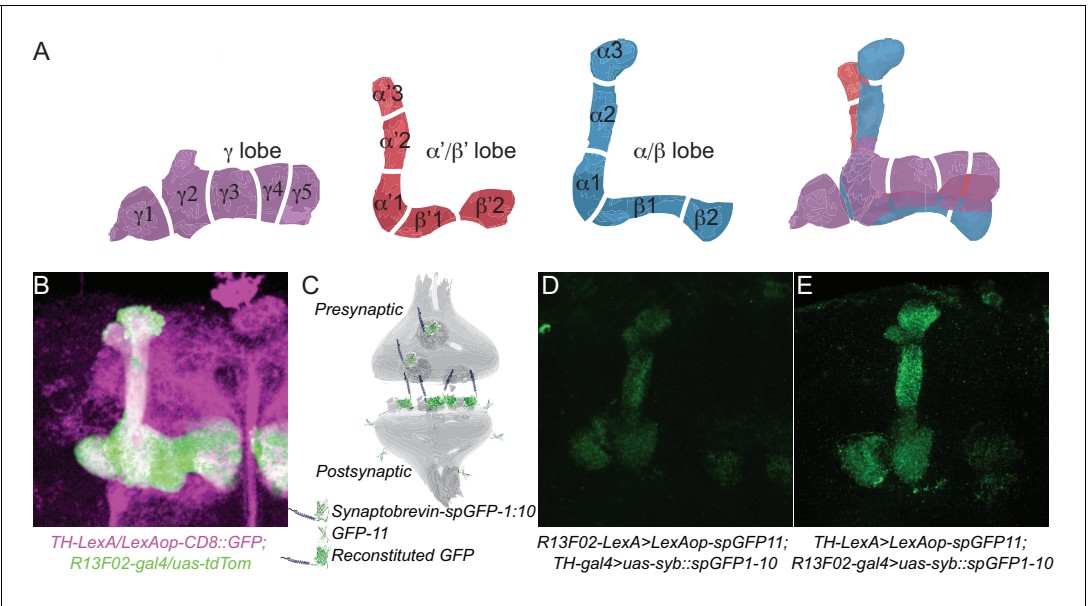

**Figure 1.** KC and DAn PPL1 neurons form axoaxonic reciprocal synapses. (**A**) Diagram of 15 discrete compartments that divide the MB lobes according to the DAn pre- and MBOn postsynaptic innervation pattern. (**B**) Co-labeling of the MB lobes using *R13F02-gal4* to drive *uas-tdtomato* expression and *TH-lexA* to drive *lexAop-CD8::GFP* expression. Lobe areas that remain primarily green receive less or no innervation from the PPL1 cluster of DAn compared to those that are white. (**C**) Diagram of trans-synaptic fluorescence reconstitution. Synaptic syb::spGFP1-10 is expressed in the presynaptic neuron along with spGFP11 in the putative postsynaptic counterpart. GFP reconstitution occurs only if synaptic connectivity exists. (**D**) GFP reconstitution when syb::spGFP1-10 is expressed in DAn. (**E**) GFP reconstitution when syb::spGFP1-10 is expressed in KC.

The following figure supplement is available for figure 1:

**Figure supplement 1.** No GFP signal was observed after staining brains that expressed only half of the GRASP system.

(***Berry et al., 2015***). As expected, expressing presynaptic split GFP (*uas-syb::spGFP1-10*) in DAn using *TH-gal4* along with membrane tagged complementary split GFP (*lexAop-CD4::spGFP11*) in KC with *R13F02-lexA* reconstituted GFP fluorescence in all the aforementioned compartments (***Figure 1D***). Surprisingly, the reciprocal experiment, with *uas-syb::spGFP1-10* expressed in the KC and *lexAop-CD4::spGFP11* in the DAn, reconstituted GFP fluorescence in an identical pattern of MB lobe compartment expression (α'3, α3, α2α'2, γ1ped, γ1, and γ2α'1; minimal fluorescence also in PAM β2β'2a) (***Figure 1E***). Three-dimensional reconstruction of images obtained by structured illumination microscopy showed well-defined synaptic puncta in the α2α'2 domain (***Video 1***). No staining was observed when brains expressing either half of the GRASP components were imaged under the same conditions (***Figure 1—figure supplement 1***). These results strongly suggest the presence of reciprocal axoaxonic synaptic connectivity between KC and the DAn represented in the TH driver with the connectivity occurring essentially across the entire vertical MB lobe and the lateral aspect of the horizontal lobe.

To show functional connectivity between KC and DAn, we used the ATP/P2X$_2$ system in ex

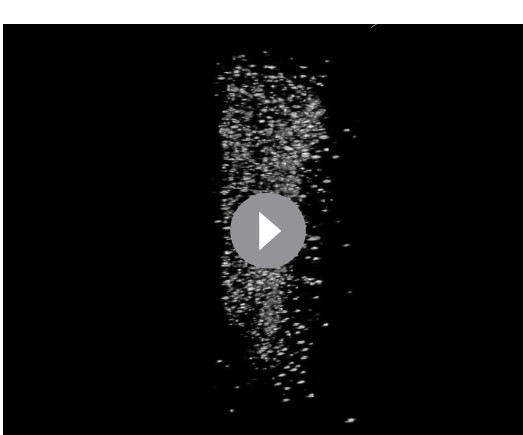

**Video 1.** A three-dimensional structured illumination image of the α2α'2 MB lobe compartment showing pre-synaptic KC puncta with post-synaptic DAn specializations on their axons.

vivo preparations to stimulate the KC and recorded cAMP and calcium influx in the DAn axon terminals using an Epac1 reporter and GCaMP3, respectively. We first verified that the ATP/P2X$_2$ system was working properly by co-expressing P2X$_2$ and the GCaMP3 reporter in DAn using *TH-gal4* and recording stimulated calcium responses to bath applied ATP. A dose-response curve to ATP confirmed the efficacy of artificially activating DAn using P2X$_2$/ATP (*Figure 2—figure supplement 1*). We then tested the pre-synaptic function of DAn and found that artificial stimulation of DAn contained in the *TH-gal4* driver induced increasing elevations of cAMP in postsynaptic KC with increasing concentrations of ATP (*Figure 2—figure supplement 2*). We failed to observe calcium responses in the KC axons when DAn were stimulated with increasing concentrations of ATP (not shown). This is consistent with prior observations (*Tomchik and Davis, 2009*) and indicates that DA receptor engagement on KC produces the modulatory signal of stimulating cAMP accumulation without altering ion channel function to allow calcium influx. When we stimulated the KC with the ATP/P2X$_2$ ion channel, we discovered potent cAMP and calcium responses in DAn (*Figure 2A,B*). Although not strictly comparable, the evoked cAMP responses in the DAn were roughly two fold larger than responses evoked in the KC with DAn stimulation (*Figure 2—figure supplement 2*). We discovered during the course of these experiments that our control genotype containing uas-P2X$_2$ alone showed significant calcium responses with ATP application (*Figure 2—figure supplement 3A*). This suggested leaky expression of the P2X$_2$ channel in PPL1 neurons despite reports that no leaky expression occurs in other types of neurons (*Yao et al., 2012*). Although responses in this control were significantly lower than the experimental group, we chose to use optogenetic tools to verify our results with the P2X$_2$ system (*Figure 2—figure supplement 3B–E*). We first verified that we could stimulate KC using Chrimson. For this, we coexpressed Chrimson along with GCaMP6 in KC and recorded calcium responses to red light stimulation in living animals using two-photon microscopy. Strong calcium responses were observed in KC to red light stimulation. Control animals without Chrimson expression showed no response (*Figure 2—figure supplement 3B*). Similar to our results with ATP/P2X$_2$, stimulating KC with red light produced potent calcium responses in PPL1 DAn (*Figure 2—figure supplement 3C*). Because the functional imaging responses and functional GRASP reconstitution experiment showed the largest responses in the α2α'2 DAn, we focused the remainder of our experiments on this particular neuron and the compartment of the MB lobes that it innervates.

To exclude the possibility that the evoked responses in the DAn were from polysynaptic innervation, we measured calcium responses in α2α'2 DAn to focal application of ATP through a glass micropipette using fast pressure injection in presence or absence of 1 µM tetradotoxin (TTX). TTX inhibits neuronal firing by binding to voltage-gated sodium channels and inhibiting action potentials driven from polysynaptic transmission. To ensure that TTX treatment was effective, we first measured calcium responses in both the KC calyx and lobes simultaneously after focal application of 10 mM ACh to the MB calyx in the presence or absence of TTX. As expected, robust calcium responses were observed in both calyx and lobes to ACh applied to the calyx. Nevertheless, the lobe but not calyx responses were completely abolished when TTX was present (*Figure 2C–D*), demonstrating that our procedures utilizing TTX effectively inhibited action potential propagation. Using this method, no difference in DAn calcium responses was observed after focal application of ATP in presence or absence of TTX (*Figure 2E–F*). Similar results were obtained using optogenetic stimulation (*Figure 2—figure supplement 3D*). These results, together with the functional trans-synaptic fluorescence reconstitution (*Figure 1C–E*) demonstrate that DAn are both pre- and post-synaptic KC, through axoaxonic reciprocal connections.

## The connection between KC and DAn is cholinergic

Recent studies revealed that KCs are cholinergic and nAChR receptors in MBOn are required for memory retrieval (*Barnstedt et al., 2016*). To explore the chemical nature of KC>DAn synaptic connection, we recorded ex vivo, ATP/P2X$_2$ evoked calcium responses in the α2α'2 DAn axon terminals in brains pre-incubated for 10 min with a battery of different inhibitors. We included inhibitors to nicotinic acetylcholine receptors (mecamylamide), muscarinic acetylcholine receptors (atropine), GABA receptors (CGP 35348), mGlu receptors (S-MCPG) and dopamine/serotonin receptors (pimozide). Only the nicotinic AChR inhibitor, mecamylamide (100 µM), significantly reduced ATP/P2X$_2$ evoked calcium responses in DAn (*Figure 3A*). We then completed a full dose-inhibition curve to increasing concentrations of mecamylamide by comparing the responses in the presence of the inhibitor to a

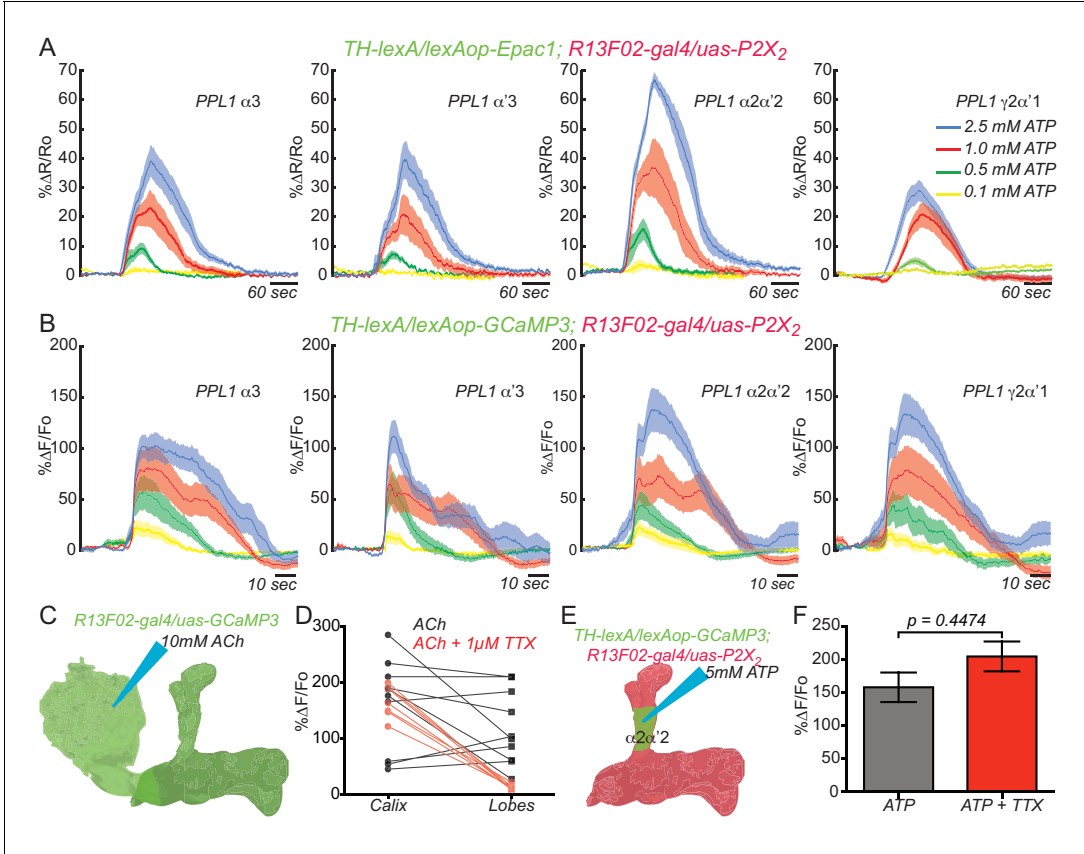

**Figure 2.** KC and DAn PPL1 neurons axoaxonic reciprocal synapses are functional. (**A**) Cyclic AMP responses in DAn (α3, α'3, α2α'2 and γ2α'1) to artificial stimulation of KC using ATP/P2X$_2$ system. The solid line of each trace represents the mean and shaded area represents ± SEM at various concentrations of ATP used in the experiment. Cyclic AMP accumulation was observed in the axons of the DAn that innervate all of the MB lobe compartments that were recorded. $N = 9–14$. (**B**) Calcium responses in DAn (α3, α'3, α2α'2 and γ2α'1) to artificial stimulation of KC using ATP/P2X$_2$ system. The solid line of each trace represents mean and shaded area represents ± SEM. Calcium responses were observed in the axons of the DAn that innervate all of the MB lobe compartments that were recorded. $N = 12–16$. (**C**) Diagram of the experimental setup. A micropipette was used to focally apply ACh to the calyx of the MB while imaging in both the calyx and MB lobes. (**D**) Calcium responses in both calyx and lobes of the KC to ACh application in the presence (red) or absence (black) of 1 μM TTX. TTX blocked the responses in the lobes but not the calyx, indicating that TTX was functional in blocking action potentials. $N = 7–10$. (**E**) Diagram of the experimental setup. A micropipette was used to focally apply 5 mM ATP to the α2α2' compartment of MB while recording calcium responses in α2α'2 DAn. (**F**) Calcium responses in α2α'2 DAn in the presence (red) or absence (gray) of 1 μM TTX. TTX was without significant effect, indicating that the artificial activation of KC fibers were capable of evoking responses in DAn axon terminals through local, mono-synaptic transmission. $N = 9–11$. Data were analyzed using Mann-Whitney U non-parametric test. Bars represent the means ± SEM.

The following figure supplements are available for figure 2:

**Figure supplement 1.** Testing the efficacy of the ATP/P2X$_2$ system.

**Figure supplement 2.** DAn are presynaptic to KC.

**Figure supplement 3.** KC and PPL1 DAn axoaxonic reciprocal synapses are functional.

second response recorded ten minutes after wash out of the inhibitor. Up to sixty percent inhibition was obtained with 250 and 500 μM mecamylamide when measured in this manner (**Figure 3B**). We suggest that full inhibition was not achieved due to the leaky expression of P2X$_2$ channel and perhaps incomplete washout. However, optogenetics experiments showed >85% inhibition with 500 μM mecamylaminde (**Figure 2—figure supplement 3E**). We then reasoned that if ACh is the neurotransmitter for the KC>DAn axoaxonic synapses, then DAn should respond directly to ACh. We

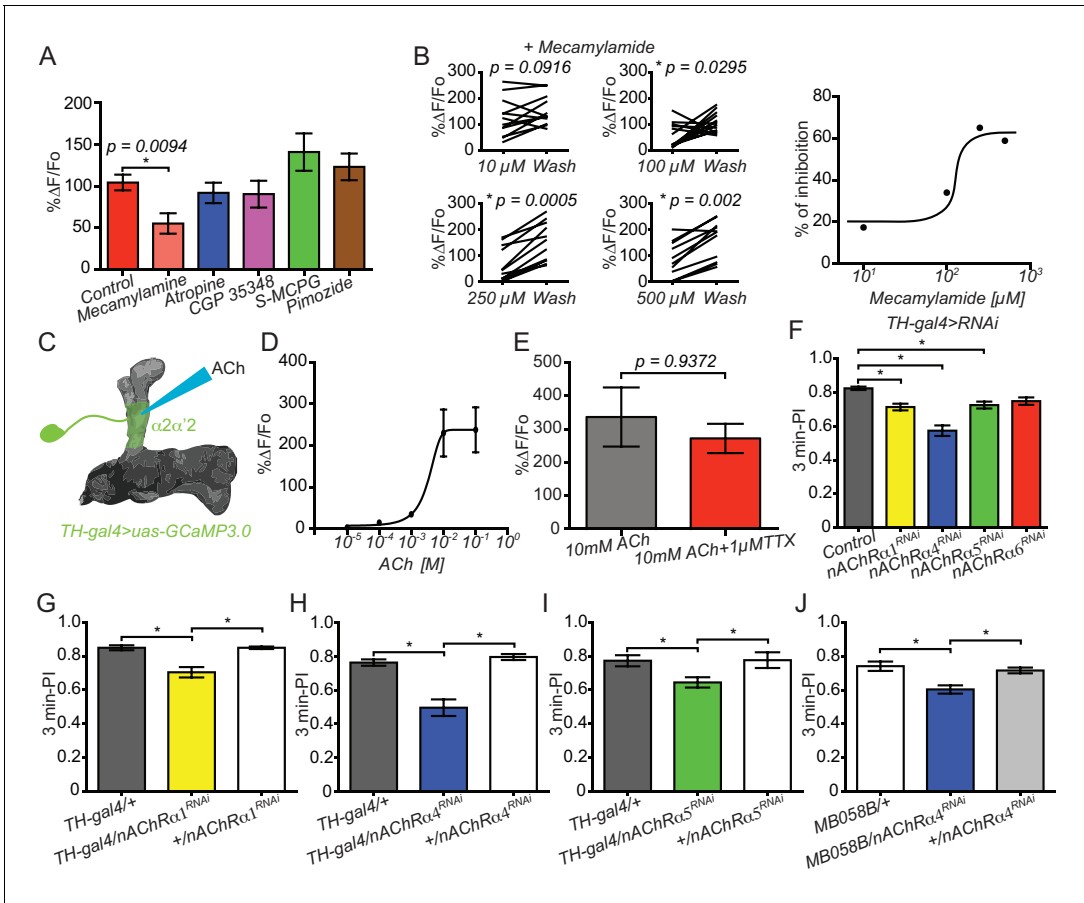

**Figure 3.** Connections between KC and DAn are cholinergic. (**A**) Calcium responses in α2α′2 DAn to stimulation of KC using the ATP/P2X$_2$ system in presence of inhibitors to nACh receptors (100 μM mecamylamide), mACh receptors (atropine), GABA receptors (CGP 35348), mGlu receptors (S-MCPG) and dopamine/serotonin receptors (pimozide). N = 10–16. Data were analyzed with Kruskal-Wallis one-way ANOVA and Dunn's multiple comparison test. Bars represent the means for calcium response ± SEM. (**B**) Dose-response plots to increasing concentrations of mecamylamide comparing the responses in the presence of the inhibitor to a second response recorded after a ten-minute washout. N = 10–16. (**C**) Diagram of the experiment setup. A micropipette was used to focally apply ACh to α2α′2 DAn axonal terminals while recording calcium responses. (**D**) Calcium dose-response curve in α2α′2 DAn terminals to focal application of increasing concentrations of ACh. N = 6. Error bars represent SEM. (**E**) Calcium responses to focal application of 10 mM ACh to α2α′2 DAn axons in the presence (red) or absence (black) of 1 μM TTX. N = 6. Data were analyzed using Mann-Whitney U non-parametric test. Bars represent the mean ± SEM. (**F**) Three-minute aversive olfactory memory of flies expressing RNAi transgenes to different nAChRα subunits in DAn using *TH-gal4*. N = 12. Data were analyzed with Kruskal-Wallis one-way ANOVA and Dunn's multiple comparison test. *p<*0.05*. Bars represent the mean ± SEM. (**G**) Three-minute aversive olfactory memory of flies expressing an RNAi transgene to *nAChRα1* RNAi compared to the parental controls, *TH-gal4/+* and *nAChRα1RNAi/+*. N = 6. Data were analyzed with Kruskal-Wallis one-way ANOVA and Dunn's multiple comparison test. Bars represent the mean ± SEM. p<0.05. (**H**) Three-minute aversive olfactory memory of flies expressing an RNAi transgene to *nAChRα4* compared to the parental controls, *TH-gal4/+* and *nAChRα4RNAi/+*. N = 6. Data were analyzed with Kruskal-Wallis one-way ANOVA and Dunn's multiple comparison test. Bars represent the mean ± SEM. p<0.05. (**I**) Three-minute aversive olfactory memory of flies expressing an RNAi transgene to *nAChRα5* compared to the parental controls, *TH-gal4/+* and *nAChRα5RNAi/+*. N = 6. Data were analyzed with Kruskal-Wallis one-way ANOVA and Dunn's multiple comparison test. Bars represent the mean ± SEM. p<0.05. (**J**) Three-minute aversive olfactory memory of flies expressing an RNAi transgene to *nAChRα4* in α2α′2 PPL1 neurons compared to the parental controls, *MB058B-split-gal4/+* and *nAChRα4RNAi/+*. N = 6. Data were analyzed with Kruskal-Wallis one-way ANOVA and Dunn's multiple comparison test. Bars represent the mean ± SEM. p<0.05.

The following figure supplement is available for figure 3:

**Figure supplement 1.** Knockdown of nAChR subunits using RNAi in DAn does not alter sleep patterns.

therefore measured calcium responses in α2α'2 DAn axon terminals to focally applied ACh. As predicted, focal application of ACh robustly increased intracellular calcium in DAn projections, reaching maximum responses at 10 and 100 mM concentrations (*Figure 3C–D*). We subsequently recorded calcium responses to focally applied ACh in the presence of 1 µM TTX to rule out the possibility that the DAn responses resulted from indirect activation of other neurons including KC. The magnitude of ACh evoked calcium responses was no different in presence or absence of TTX, indicating that these neurons have ACh postsynaptic specializations in their axonal terminals located in the MB neuropil (*Figure 3E*).

## Cholinergic input to DAn is required for normal olfactory learning

As described above, the PPL1 DAn have been widely modeled as conveying US shock information to the KC during olfactory conditioning. Because of this, we probed the possible role of the KC⇋DAn reciprocal connection for this behavior by reducing the ACh input to the DAn axons from the KC. To do so, we knocked down several nAChR subunits using multiple RNAi transgenes and tested the flies for performance after olfactory conditioning. Flies expressing RNAi to nAChRα1, nAChRα4, and nAChRα5 but not nAChRα6 using *TH-gal4* modestly impaired aversive olfactory learning when compared to control flies. Control flies were constructed by crossing *TH-gal4*-containing flies to the RNAi parental line 36303, which contains an empty docking site, *attp40* (*Figure 3F*). This *attp40* site was used to insert RNAi constructs to obtain flies harboring the RNAi-transgenes. Similar results were obtained when flies expressing RNAi's to nAChRα1, nAChRα4, and nAChRα5 were individually compared to their two parental genetic controls, *TH-gal4/+* and *uas-RNAi/+* (*Figure 3G–I*), suggesting that these three subunits are expressed on the axons of DAn. Similar results were observed when we used a Split-gal4 driver, MB058B, to restrict nAChRα4 KD in α'2 α'2 PPL1 neurons. No significant differences were observed in olfactory and electric shock avoidance for any of the genotypes used for behavioral experiments in our study (data not shown). Importantly, these results coupled with those showing KC axoaxonic synapses to DAn indicate that KC-driven cholinergic input to the distal axons of DAn is required for normal learning, possibly due to amplification of dopamine release at acquisition.

Since the DAn>KC circuit is known to be involved in sleep and sleep homeostasis, we also evaluated sleep profiles in flies expressing RNAi transgenes to nAChR subunits in DAn. We found no significant differences in sleep patterns compared to control flies (*Figure 3—figure supplement 1*).

## KC input to DAn is required for normal DAn synaptic release during learning

If the cholinergic input from KC to DAn is required for full DA release during learning as suggested by our results above, then blocking MB output should impair DAn synaptic release during memory acquisition. To test this prediction, we silenced KC using Kir2.1 channel expression while recording presynaptic calcium responses using syt:GCaMP6s during a training session in the α2α'2 PPL1 DAn axon terminals. This reporter uses the genetically encoded calcium sensor GCaMP6s tethered to the C terminus of the vesicular synaptic protein synaptotagmin. Presynaptic calcium is a key regulator of neurotransmission and this sensor provides a sensitive readout of synaptic release (*Cohn et al., 2015*). Flies of the genotype *R82C10-LexA/lexAop-sytGCaMP6s; R13F02-gal4/uas-kir2.1, tub-gal80^{ts}* were reared at 18°C and then switched at eclosion to 30°C for 16 hr to allow expression of *kir2.1* in KC (*Cervantes-Sandoval et al., 2013*) (*Figure 4A*). Expression of Kir 2.1, an inwardly rectifying potassium channel, silences neurons by inhibiting membrane depolarization. Presynaptic calcium responses were then recorded in flies subjected to aversive olfactory training under the microscope at room temperature. The responses were then compared to flies where *kir2.1* expression was not induced by maintaining the flies at 18°C. A significant decrease in presynaptic calcium responses was observed when KC were silenced with *kir2.1* compared to flies without *kir2.1* expression (*Figure 4B*). The shape of α2α'2 PPL1 responses during CS/US conditioning suggested that they are bipartite, consisting of a rapidly adapting odor response with superimposed electric shock peaks. To determine which of the two components require KC input, we recorded responses in α2α'2 DAn with odor or shock delivery separately. Surprisingly, we found that silencing KC impaired both odor- and shock-induced presynaptic calcium responses in α2α'2 DAn (*Figure 4C,D*). As expected, control flies without the *kir2.1* transgene (*R82C10-LexA/lexAop-sytGCaMP6s; R13F02-gal4/tub-gal80^{ts}*) did not

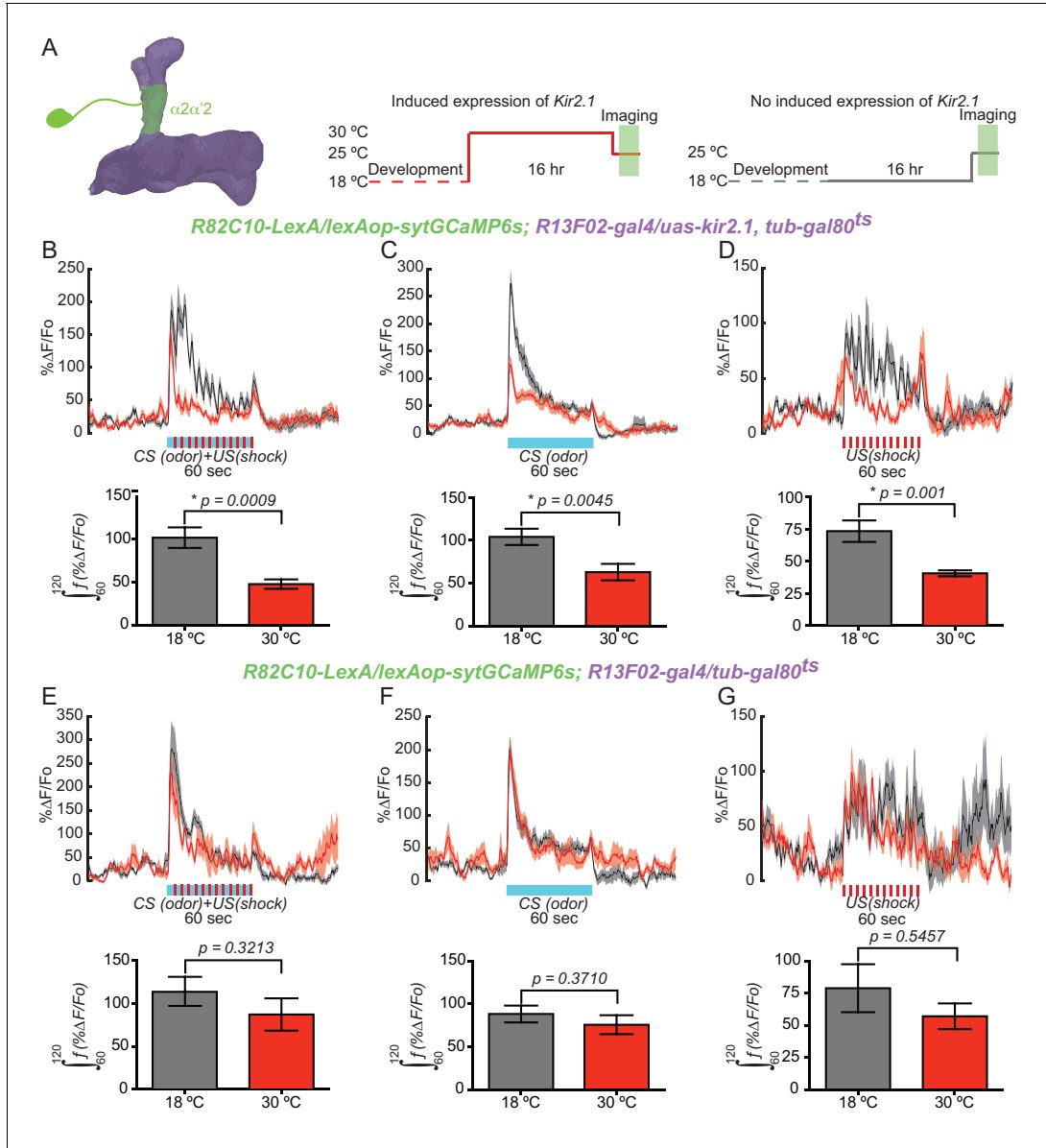

**Figure 4.** KC input is required for normal DAn synaptic release during learning. (**A**) Diagram of the experimental setup. Flies of the indicated genotype were reared at 18°C and then 1–2 day old adults were switched to 30°C to induce *kir2.1* channel expression in KC for 16 hr. Flies were then returned to room temperature and prepared for functional imaging. Control flies remained at 18°C until removing them to room temperature for functional imaging. (**B**) Calcium responses in α2α′2 DAn during aversive olfactory training (odor+shock) in flies with (red) or without (black) *kir2.1* expression in KC. $N$ = 10–12. The solid line represents the mean and the shaded area represents ± SEM. Data were analyzed using Mann-Whitney U non-parametric test. Bars represent the mean ± SEM of the area under the curve during the one minute of conditioning (the 60–120 s imaging time window). (**C**) Calcium responses in α2α′2 DAn during odor delivery in flies with (red) or without (black) *kir2*.1 expression in KC. $N$ = 13–14. The solid line represents the mean and the shaded area represents ± SEM. Data were analyzed using Mann-Whitney U non-parametric test. Bars represent the mean ± SEM of the area under the curve during the one minute of odor presentation (the 60–120 s imaging time windows). (**D**) Calcium responses in α2α′2 DAn during 12 shock pulses delivery in flies with (red) or without (black) *kir2.1* expression in KC. $N$ = 14. The solid line represents the mean and the shaded area represents ± SEM. Data were analyzed using Mann-Whitney U non-parametric test. Bars represent the mean ± SEM of the area under the curve during the one minute of shock pulses (the 60–120 s imaging time window) (**E–G**) Parallel data to panels **B–D** but using flies without the *kir2.1* transgene. These control data show that the differential responses observed in panels **B–D** require the *kir2.1* expression.

show significant differences in presynaptic calcium responses to conditioning, odor or shock presentation (*Figure 4E–G*). These results show that normal presynaptic calcium responses and dopamine release requires KC input to DAn during conditioning and with odor or shock stimuli presented alone. Most importantly, these results are consistent with a model of cholinergic input from KC representing a positive gain signal to DAn for subsequent DA release.

## KC input shapes DAn ongoing activity

DAn respond to a variety of external and internal stimuli (*Riemensperger et al., 2005*; *Mao and Davis, 2009*; *Tomchik, 2013*; *Liu et al., 2012*; *Lin et al., 2014b*; *Lewis et al., 2015*; *Musso et al., 2015*) and two recent studies have shown that ongoing dopaminergic activity is correlated with the behavioral state of the animal (*Berry et al., 2015*; *Cohn et al., 2015*). These observations suggest that the ongoing activity of the distributed DAn network represents the current external, internal and behavioral state of the animal providing for a moment-by-moment update of these elements in the fly (*Berry et al., 2015*; *Cohn et al., 2015*; *Waddell, 2016*). This hypothesis emphasizes the need to understand how the ongoing DAn activity is generated. Using the same tools as above, we tested the role for KC by recording ongoing calcium transients of α2α'2 PPL1 neurons for 10 min while silencing KC with expression of *kir2.1* (*Figure 5A*). Ongoing activity recorded in the α2α'2 PPL1 neuron axon terminals with the syt:GCaMP6s presynaptic reporter showed robust calcium transients compared to the previously reported lack of activity recorded with GCaMP3 (*Berry et al., 2012*, *Berry et al., 2015*; *Cervantes-Sandoval et al., 2016*) (*Figure 5B*). Silencing KC significantly reduced total ongoing activity in α2α'2 PPL1 neuron axon terminals compared to flies without *kir2.1* expression (*Figure 5B–C*). Upon quantification, we found no change in the number of peaks counted between the experimental and control groups (*Figure 5D*). Nevertheless, a significant decrease was observed in the magnitude of these peaks when KCs were silenced with *kir2.1* (*Figure 5E*). As expected, control flies without the *kir2.1* transgene (*R82C10-LexA/lexAop-sytGCaMP6s; R13F02-gal4/tub-gal80^{ts}*) did not show significant differences in total ongoing DAn activity, peak number or peak magnitude (*Figure 5F–I*), indicating that the differences observed with flies containing *kir2.1* were not attributable to the temperature differences in the experimental protocol (*Figure 5A*). These results indicate that KC input to DAn shapes the magnitude of this chronic activity although it may not generate it. In addition, our results demonstrate that both phasic/evoked and chronic/ongoing activity of DAn is influenced by direct cholinergic input from KC through axoaxonic reciprocal synapses.

## Concluding remarks

Our results demonstrate that DAn are both pre- and post-synaptic to KC through axoaxonic reciprocal connections, in contrast to current models which envision them as providing only pre-synaptic input. The DAn>KC half of the reciprocal synapse employs DA as neurotransmitter, although we cannot rule out the possibility that other neurotransmitters are co-released with DA. The KC>DAn half of the reciprocal synapse is cholinergic. However, the fact that we were able to observe both cAMP and calcium responses in DAn with KC stimulation suggests that there may be other mediators of this reciprocal connection. Blocking the cholinergic input to DAn attenuates aversive olfactory learning, providing evidence that its function is, at least in part, to provide an amplification signal for the initial DA release due to activating the US pathway. Consistent with this role we found that silencing KC impairs DAn presynaptic calcium responses to conditioning, odor and shock stimuli, presumably influencing dopamine release and explaining the learning phenotype. Overall, our results support the existence of a positive feedback loop (*Riemensperger et al., 2005*) required for optimal learning. We envision that DAn receive direct input from the US during conditioning which is conveyed to KC. The KCs also receive coincident olfactory input and this coincidence provides positive feedback onto the DAn through cholinergic synapses to further increase DAn activity.

Our results also show that KC input to DAn shapes their ongoing or chronic activity. It is plausible that ongoing activity in the DAn provides a moment-by-moment update of the external environment and internal states and the behavioral status of the fly that appropriately reconfigures the KC>MBOn flow of information (*Berry et al., 2015*; *Cohn et al., 2015*; *Waddell, 2016*). Thus, the DAn/KC/MBOn circuit may form a recurrent network that serves as the insect's brain center for the rapid integration of sensory information and decision-making (*Lewis et al., 2015*). Local feedback loops,

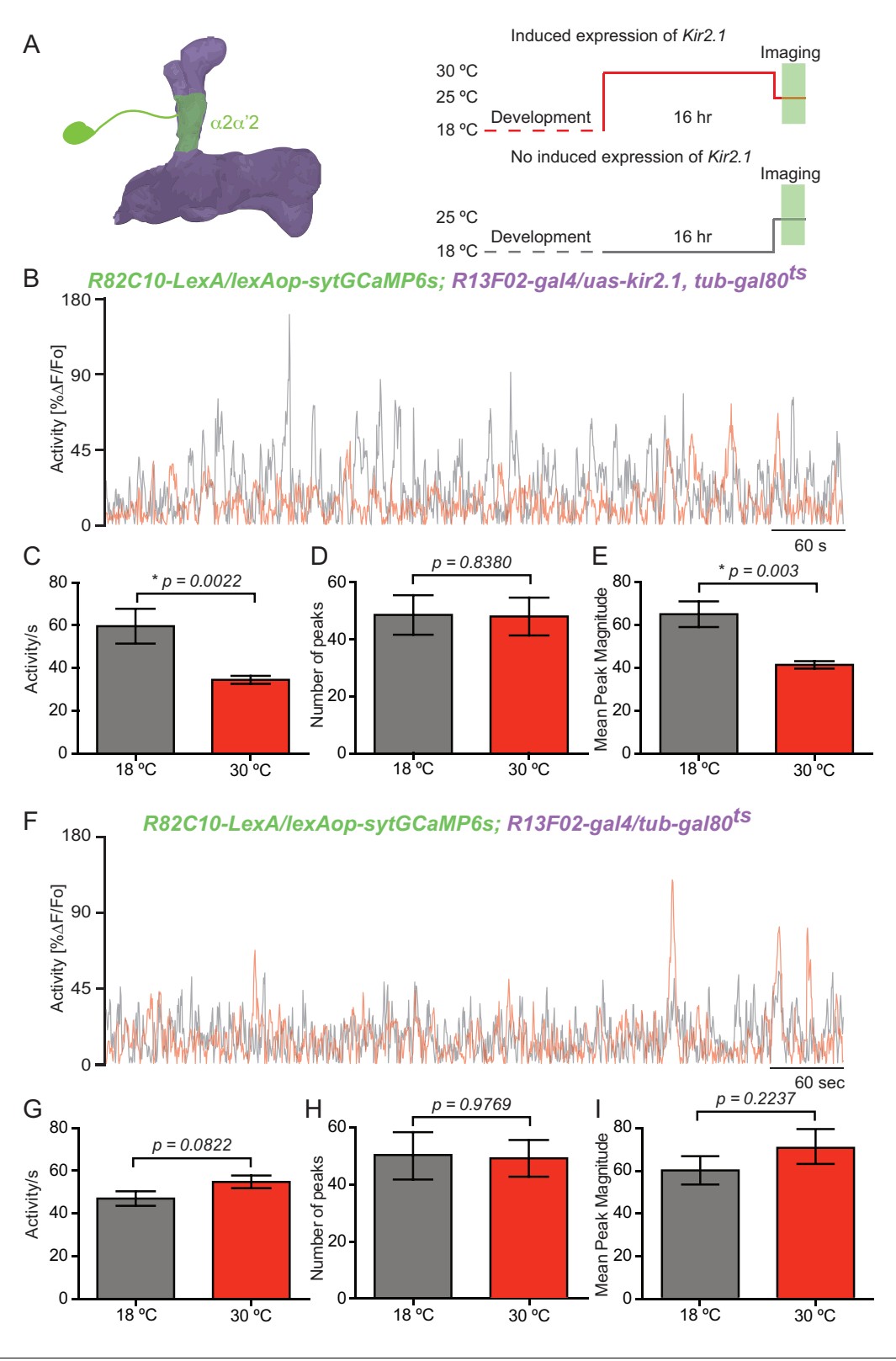

**Figure 5.** KC input shapes DAn ongoing activity. (**A**) Diagram of the experimental setup. Flies of the indicated genotype were reared at 18°C and 1–2 day old adults were then switched to 30°C to induce *kir2.1* channel expression in KC for 16 hr. The flies were then returned to room temperature and prepared for functional imaging. Control flies remained at 18°C until removing them to room temperature for functional imaging. (**B**) Representative

*Figure 5 continued*

10 min recording of calcium ongoing activity in α2α'2 DA axon terminals in flies with (red) or without (black) *kir2.1* expression in the KC. (**C**) Mean of total activity per sec during 10 min of recording of ongoing activity in α2α'2 DAn axon terminals in flies with (red) or without (black) *kir2.1* expression in the KC. *N = 10–12*. Data were analyzed using Mann-Whitney U non-parametric test. Bars represent the mean ± SEM. (**D**) Mean number of peaks of the same 10 min recordings of ongoing activity in α2α'2 DAn axon terminals in flies with (red) or without (black) *kir2.1* expression in the KC. Data were analyzed using Mann-Whitney U non-parametric test. Bars represent the mean ± SEM. (**E**) Mean peak magnitude of ongoing activity in α2α'2 DAn in flies with (red) or without (black) *kir2.1* expression in the KC. Data were analyzed using Mann-Whitney U non-parametric test. Bars represent the mean ± SEM. (**F–I**) Parallel data to panels **B**–**E** but using flies without the *kir2.1* transgene. These control data show that the decrease in total activity and mean peak magnitude observed in panels **B** and **E** require the *kir2.1* expression. *N = 11–13*. Data were analyzed using Mann-Whitney U non-parametric test.

achieved by reciprocal connectivity like that described here, may provide computational benefits to fine tune and optimize the output. Behavioral flexibility may be achieved by passing information through local reentrant loops with constant updating from the external or internal state of the organism (*Person and Khodakhah, 2016*).

A caveat in this and other studies (*Barnstedt et al., 2016*) is that we cannot exclude that other, non-KC, cholinergic input to DAn contributes to memory acquisition. Massive efforts to generate 'connectomes' in multiple species (*Bargmann and Marder, 2013*) may offer resolution to this issue at some point in the future. Alternatively, they may continue to reveal additional connections and complexities that defy an immediate understanding. Studies like the present one, that reveal unexpected relationships between synaptic partners in difficult-to-untangle circuits, expose the need to advance beyond 'connectomics' and develop new tools that allow us to silence or activate specific channels and specific synaptic connections between neurons of interest without affecting other functions in the same cells (*Bargmann and Marder, 2013*).

## Materials and methods

### Drosophila husbandry
Flies were cultured on standard medium at room temperature. Crosses, unless otherwise stated, were kept at 25°C and 70% relative humidity with a 12 hr light-dark cycle. The Gal4 drivers used in this study include *R13F02-gal4, R82C10-lexA* (*Jenett et al., 2012*), *TH-gal4* (*Friggi-Grelin et al., 2003*), and *TH-lexA* (*Berry et al., 2015*). The *uas*-RNAi transgene stocks were obtained from the TRiP collection library (*Perkins et al., 2015*) and included control line (36303), which contains the *attP* docking site without an RNAi insertion (hence same insertional modification as *uas-RNAi* lines), and the *nAChR^RNAi* lines (28688, 31985, 25943, 25835). Additional transgene stocks included *uas-dicer2, lexAop-GFP::CD8, uas-tdTomato* (*Pfeiffer et al., 2010*), *uas-syb::spGFP1-10, lexAop-CD4::spGFP11* (*Macpherson et al., 2015*), *lexAop-epac1, uas-epac1, lexAop-P2X2, uas-P2X2, lexAop-GCamp3.0* (*Lima and Miesenböck, 2005*; *Yao et al., 2012*), *uas-GCaMP3* (*Tian et al., 2009*), *lexAop-syt::GCaMP6s* (*Cohn et al., 2015*), *uas-GCaMP6f* (*Chen et al., 2013*), *uas-CsChrimson::TdTomato, lexAop-CsChrimson::TdTomato* (*Hoopfer et al., 2015*), *uas-kir2.1* (*Baines et al., 2001*), and *tub-gal80^ts* (*McGuire et al., 2003*).

### Immunostaining
Whole brains were isolated and processed with minor modifications of those described (*Jenett et al., 2012*). Brains were first incubated with primary antibodies including: rabbit polyclonal anti-DsRed(1:1000, Clontech Labs, cat# 632496, RRID:AB_10013483) and mouse monoclonal anti-GFP (1:1000, Sigma, cat# G6539; anti-GRASP, RRID:AB_259941). Secondary antibodies included: anti-rabbit IgG conjugated to Alexa Fluor 488 (1:800, Life Technologies Cat# A11008, RRID: AB_143165) and anti-mouse IgG conjugated to Alexa Fluor 633 (1:400, Life Technologies Cat# A21052, RRID: AB_141459). Images were collected using a 10X objective with a Leica TCS SP5 II confocal microscope with 488 and 633 nm laser excitation.

For structured illumination microscopy (SIM), brains were processed as for standard immunostaining plus these additional steps: After the wash steps following secondary antibody incubation, brains were post-fixed in 1%PFA in PAT3 (0.5% BSA, 0.5% Triton, 1X PBS) for two hr at room temperature. The brains were then washed for 2 hr in 1 mL of PAT3 buffer two times, followed by a quick wash with 1X PBS and then one with dH$_2$O. The brains were mounted onto high precision 18 × 18 mm coverslips previously coated with poly-lysine with the anterior face towards the coverslip for imaging the MB lobes. Excess moisture was removed from the coverslips, and these were then dehydrated by 5 min incubations in a series of ethanol dilutions (20%, 30%, 50%, 70%, 95%, 100%x2). The brains were cleared with an overnight incubation in methyl salicylate and then the brains with the coverslip still attached were sandwiched with a second coverslip between two pieces of plastic. The coverslips were then sealed with nail polish. SIM images were collected on Zeiss ELYRA PS1 and examined using ImageJ.

## Functional imaging experiments

### Bath applied experiments

These experiments were performed as described previously with minor modifications (*Tomchik and Davis, 2009*). Briefly, brains were dissected at room temperature and maintained in continuous perfusion of saline solution (2 mL/min; 103 mM NaCl, 3 mM KCl, 5 mM HEPES, 1.5 mM CaCl$_2$, MgCl$_2$, 26 mM NaHCO$_3$, 1 mM NaH$_2$PO$_4$, 10 mM trehalose, 7 mM sucrose, and 10 mM glucose [pH 7.2]) under a confocal microscope. Adenosine triphosphate (ATP) at different concentrations was applied for 30 s in perfusion solution and then subsequently washed out. Recordings were collected at 2 frames/sec during a 10 min recording session. Responses were plotted as the baseline-normalized change in GCaMP fluorescence (%$\Delta$F/Fo) or change in epac1-camps inverse FRET ratio (%$\Delta$R/Ro) within a defined region of interest.

### Focal application experiments

For focal application of compounds, brains were dissected and treated for 30 s with 2 mg/mL collagenase and washed in saline for 1 min before initiating the recording. Acetylcholine or ATP was dissolved in saline with 5 mM Texas red dextran (to allow optical monitoring of the stimulus). The solutions were applied focally to the α2α'2 MB lobe compartment or calyx via pressure ejection from a glass micropipette (~5 μm tip diameter). The glass micropipette was previously coated with BSA-conjugated Texas Red (Life Technologies, Grand Island, NY) in order to visualize the micropipette tip position.

### In vivo imaging

For measuring calcium responses with conditioning, odor or shock delivery, we processed flies as previously described (*Tomchik and Davis, 2009*). Briefly, a single fly was aspirated without anesthesia into a narrow slot the width of a fly in a custom-designed recording chamber. The head was immobilized by gluing the eyes to the chamber using myristic acid and the proboscis fixed to reduce movements. A small, square section of dorsal cuticle was removed from the head to allow optical access to the brain. Fresh saline (103 mM NaCl, 3 mM KCl, 5 mM HEPES, 1.5 mM CaCl$_2$, MgCl$_2$, 26 mM NaHCO$_3$, 1 mM NaH$_2$PO$_4$, 10 mM trehalose, 7 mM sucrose, and 10 mM glucose [pH 7.2]) was perfused immediately across the brain to prevent desiccation and ensure the health of the fly. Using a 25X water-immersion objective and a Leica TCS SP5 II confocal microscope with a 488 nm argon laser, we imaged the α2α'2 neuron for 3 min, during which stimuli (odor+shock, odor or shock) was delivered starting at the 2nd min. We used one PMT channel (510–550 nm) to detect syt::GCaMP6s fluorescence. We set the baseline for α2α'2 ongoing activity detected with syt::GCaMP6s as the mean of minimums across 10 s bins of the first 60 s of the recording. This baseline was then used to calculate %$\Delta$F/F$_o$ for the complete recording. Total activity was quantified as the area under the curve during the 1 min corresponding to stimulus delivery (between 60 and −120 sec of the recording). Expression of *kir2.1* was induced by transferring flies to 30°C for 16 hr. Flies were then returned to room temperature and imaged for the following 3 hr.

### Ongoing activity recording

Ongoing activity of α2α′2 was recorded as previously reported (*Berry et al., 2012*). Briefly, flies processed as in the previous experiment were recorded at 2 frames/sec during a 10 min time window. The activity per sec was calculated for each recording using algorithms previously described (*Berry et al., 2012*). Peaks in these recordings were quantified using the Matlab (RRID:SCR_001622) algorithm *peakfinder* developed by *Yoder, 2011*.

## Optogenetic stimulation and two-photon in vivo calcium imaging

Combined in vivo optogenetic and calcium imaging were conducted using a two-photon microscope to minimize non-specific optogenetic simulation during recording. Flies for these experiments were collected from crosses were kept in dark on standard corn meal supplemented with 0.2 mM all trans-retinal. After eclosion, adult flies were kept for 2–5 days in food supplemented with 0.5 mM all trans-retinal. Live flies were fixed to the recording chambers as in the previous experiments (above). For light stimulation, we used a high power 617 nm LED (Red-Orange Luxeon Rebel LED-122 lm) and a 0.5 mm diameter optic fiber (Edmund Optics) placed adjacent to the eyes of a fly in a recording chamber. The light power at the fly head was measured to be 1 mW/mm$^2$. The LED was controlled using an arduino microcontroller (Arduino Uno). Light pulses were delivered at 40 Hz, with 10 ms duration for a total of 200 ms (*Barnstedt et al., 2016*).

## Behavior

Two to six-day-old flies were used for all behavior experiments. Standard aversive olfactory conditioning experiments were performed as described (*Beck et al., 2000*). Briefly, a group of ~60 flies were loaded into a training tube where they received the following sequence of stimuli: 30 s of air, 1 min of an odor paired with 12 pulses of 90V electric shock (CS+), 30 s of air, 1 min of a second odor with no electric shock pulses (CS−), and finally 30 s of air. We used 3-octanol (OCT) and 4-methylcyclohexanol (MCH) as standard odorants. To measure memory, we transferred the flies into a T-maze where they were allowed 2 min to choose between the two odors used for training.

## Monitoring sleep

*Drosophila* sleep and activity were measured using the *Drosophila* Activity Monitoring System (Trikinetcs) as described previously (*Shaw et al., 2000*). In summary, flies were placed into individual 65 mm tubes and the activity was continuously monitored. Locomotor activity was measured in 1 min bins and sleep was defined as periods of quiescence lasting at least 5 m. Sleep in min/hr was plotted as a function of circadian time (hr).

## Statistical analyses

Statistics were performed using Prism 5 (Graphpad). All tests were two-tailed and confidence levels were set at α=0.05 The figure legends present the *p* values and comparisons made for each experiment. Unless otherwise stated, non-parametric tests were used for all imaging data and DAM system monitoring, while parametric tests were used for olfactory memory comparisons as PI values are normally distributed (*Walkinshaw et al., 2015*).

## Acknowledgements

This work was supported by grants 4R37NS19904, 5R01NS052351 and 1R35NS097224 from the NINDS to RLD. We are grateful to the Iris and Junming Le Foundation for funds to purchase a super-resolution microscope. We thank Dr. Orie Shafer for sharing flies for ATP/P2X2 experiments. We thank Dr. Marco Gallio for sharing functional GRASP flies and Dr. Vanessa Ruta for sharing syt:: GCaMP6s flies.

# Additional information

## Funding

| Funder | Grant reference number | Author |
|---|---|---|
| National Institute of Neurological Disorders and Stroke | 4R37NS19904 | Ronald L Davis |
| National Institute of Neurological Disorders and Stroke | 5R01NS052351 | Ronald L Davis |
| National Institute of Neurological Disorders and Stroke | 1R35NS097224 | Ronald L Davis |

The funders had no role in study design, data collection and interpretation, or the decision to submit the work for publication.

## Author contributions

IC-S, Conceptualization, Data curation, Formal analysis, Validation, Investigation, Visualization, Methodology, Writing—original draft, Project administration, Writing—review and editing; AP, Methodology, Writing—review and editing, Developed Super-resolution imaging for whole mount *Drosophila* brains; MC, Data curation, Formal analysis, Methodology, performed all immunostaining and curation/analysis of the corresponding data; RLD, Resources, Formal analysis, Supervision, Funding acquisition, Writing—original draft, Project administration, Writing—review and editing

## Author ORCIDs

Isaac Cervantes-Sandoval, http://orcid.org/0000-0002-6372-7288
Ronald L Davis, http://orcid.org/0000-0002-5986-7608

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
