## [Decision Letter]

Thank you for submitting your article "Reciprocal synapses between mushroom body and dopamine neurons form a positive feedback loop required for learning" for consideration by *eLife*. Your article has been reviewed by two peer reviewers, and the evaluation has been overseen by K VijayRaghavan as the Reviewing Editor and Senior Editor. The reviewers have opted to remain anonymous.

The reviewers have discussed the reviews with one another and the Reviewing Editor has drafted this decision to help you prepare a revised submission.

Summary:

Isaac Cervantes-Sandoval and colleagues report experiments that demonstrate reciprocal synapses between dopaminergic neurons and intrinsic mushroom body neurons (Kenyon cells) of the *Drosophila* brain. The entire study is of importance in the context of the dissection of neuronal circuits underlying associative learning, and it is written with this emphasis. Sensory stimuli, e.g., odors, are represented in the mushroom body in terms of sparsely activated ensembles of Kenyon cells. Dopaminergic neurons synapsing onto Kenyon cell axons have been shown to convey the reinforcing properties of salient, unconditioned stimuli. A current concept postulates that coincident activity of both modulates synaptic output from Kenyon cells onto mushroom body output neurons, ultimately influencing the animals' behavior.

Here, the authors report an additional and important finding, i.e., a reciprocal synaptic connection between dopaminergic neurons and Kenyon cells. They conclude from their results that synaptic output from Kenyon cells onto dopaminergic neurons is at least partially cholinergic, that this type of synaptic transmission is required for proper associative learning, and enhances acute, stimulus-evoked as well as continuous activity in dopaminergic neurons. They propose an excitatory feedback loop between dopamine neurons and Kenyon cell that constantly updates the efficiency of dopaminergic transmission onto Kenyon cells. They base their result by experiments using a number of state-of-the-art techniques, i.e., functional reconstitution of GFP, optogenetic activation of Kenyon cells together with functional cAMP and calcium imaging in dopaminergic neurons, pharmacological and genetic (RNAi) experiments to impair cholinergic transmission, and by reducing membrane depolarization in Kenyon cells together with stimulus-evoked calcium imaging in dopaminergic neurons.

Overall, we are enthusiastic about this report, and definitely recommend publication in *eLife*. It will certainly attract the attention of readers within the large community of scientists working on *Drosophila* neuroscience. Moreover, it widens our knowledge about fundamental principles underlying the neuronal computation of associative learning in general, which will be of interest for scientists working on vertebrate brains as well. The experimental procedures, the statistical analysis or the presentation of the results, all of which are accurate and sound. The manuscript is overall clearly written, and the illustrations and well understandable. There are some significant comments to be addressed, given below.

Essential revisions:

1) Subsection “Cholinergic input to DAn is required for normal olfactory learning”, first paragraph: the authors state that the functional results with TH-Gal4 and UAS-RNAi transgenes indicate that ACh receptors are expressed on the axons on DAn.

The authors should either show directly that ACh receptors are expressed in the expected sites or provide a reference that shows this directly.

The authors should acknowledge and discuss explicitly the possibility that the learning phenotype observed in Figure 3 is influenced by an effect of reduction in ACh receptor function elsewhere, since the TH-Gal4 driver shows a broad expression pattern in Figure 1.

2) The labeling in Figure 1 is not strong. The authors should show control labeling levels observed when spGFP11 and spGFP1-10 are expressed singly.

Other comments:

1) The terminology is hard for the non-specialist to follow in some places. The use of Greek letters and prime designations in Figure 1 needs to be checked. Also, Figure 1 shows "gamma1" but not "gamma1ped". Where is PAMB2B'2a? The expression patterns of some of the drivers should be described more explicitly in the text (e.g. subsection “MBn are both pre- and post- synaptic to DAn forming axoaxonic reciprocal connections”, first paragraph).

2) Figure 1 legend: "GFP reconstitution occurs only if functional connectivity exists." This may be an overstatement; it would seem that GFP reconstitution could in principle occur if structural but not functional connectivity exists. A reference is needed if this statement is to be included.

3) Subsection “MBn are both pre- and post- synaptic to DAn forming axoaxonic reciprocal connections”, second paragraph: Please clarify why cAMP but not Ca^2+^ levels are elevated in MBn when DNa is stimulated.

4) It would be helpful to provide some general information about axonal-axonal reciprocal synapses in the Introduction, or at least to cite a reference, possibly a review article.

---

## [Author Response]

*Essential revisions:*

*1) Subsection “Cholinergic input to DAn is required for normal olfactory learning”, first paragraph: the authors state that the functional results with TH-Gal4 and UAS-RNAi transgenes indicate that ACh receptors are expressed on the axons on DAn.*

The authors should either show directly that ACh receptors are expressed in the expected sites or provide a reference that shows this directly.

Our conclusion is based on three lines of evidence. First, stimulation of cholingeric KC in the presence of TTX shows that the connectivity from KC to DAn is monosynaptic. This indicates that cholinergic receptors are expressed on the axons of DAn, since this is the only known point of direct contact between these two cell types. Second, focal application of ACh to the axons of DAn provoke strong calcium responses even in the presence of TTX, eliminating the possibility of inadvertent stimulation of other neurons. This experiment also indicates that nAChR are expressed on the axons of DAn. Third, RNAi knockdown of nACh receptor subunits impairs memory formation, using two different DAn drivers. This latter observation does not address the spatial specificity component of our conclusion but indicates that the nACh receptor is expressed in these neurons. Overall, these three lines of evidence together offer extremely strong support for our conclusion.

The reviewer’s request for us to demonstrate the presence of ACh receptors on the axons of the DAn is technically an extremely demanding one, so it is not surprising that no reference for this currently exists. It would require a tagged-ACh receptor fusion protein that one would be confident is trafficked in proper ways in an overexpression situation (Gal4 driven UAS), or high quality antibodies against the transmembrane receptor subunits for use in immunocytochemistry experiments. We did attempt the latter experiments but unfortunately all of the antibodies tested proved ineffective for immunostaining. Nevertheless, we argue that the three lines of evidence listed above offer extremely strong support for our conclusion. In the interest of a bit more caution, we changed our statement from “[…]indicating that ACh receptors are expressed on the axons on DAn” to “[…]suggesting that ACh receptors are expressed on the axons on DAn.”

*The authors should acknowledge and discuss explicitly the possibility that the learning phenotype observed in Figure 3 is influenced by an effect of reduction in ACh receptor function elsewhere, since the TH-Gal4 driver shows a broad expression pattern in Figure 1.*

To exclude the possibility that the learning phenotype observed with TH-Gal4 is due to expression of iRNA in cells other than DAn we included a new experiment in which we knocked down nAChR alpha4 only in PPL1 α2α’2 neuron with the highly specific MB058B split-gal4.

*2) The labeling in Figure 1 is not strong. The authors should show control labeling levels observed when spGFP11 and spGFP1-10 are expressed singly.*

We show the level of labeling with spGFP11 and syb::spGFP1-10 when expressed singly in Figure 1—figure supplement 1. No fluorescence was observed by each half of the split GFP molecule when imaged under the same conditions that produced detectable labeling when both were expressed together. The monoclonal antibody used in these experiments can recognize spGFP1-10 and signal is observed if the gain is turned to maximum during image acquisition. In addition, we have found that naïve fluorescence is observed in the brain only when GFP is in the reconstituted state prior to antibody staining (data not shown).

*Other comments:*

*1) The terminology is hard for the non-specialist to follow in some places.*

We have tried to simplify terminology.

*The use of Greek letters and prime designations in Figure 1 needs to be checked. Also, Figure 1 shows "gamma1" but not "gamma1ped". Where is PAMB2B'2a?*

We have checked these. Differences in form of letters and superscripts may be a function of the use of specific word processors and image construction programs. We wanted to show in Figure 1 the 15 discrete compartments into which MB are divided rather than the innervation pattern of the DAn. This is why gamma1 ped is not shown. PAMB2B’2 is clearly observed in Figure 1 and we also added this in the text.

*The expression patterns of some of the drivers should be described more explicitly in the text (e.g. subsection “MBn are both pre- and post- synaptic to DAn forming axoaxonic reciprocal connections”, first paragraph).*

We added a statement to indicate that R13F02-gal4 is quite specific to KC. We did describe where the TH-lexA driver is expressed and we also included the reference to the paper that originally described the driver (Berry et al., 2015).

*2) Figure 1 legend: "GFP reconstitution occurs only if functional connectivity exists." This may be an overstatement; it would seem that GFP reconstitution could in principle occur if structural but not functional connectivity exists. A reference is needed if this statement is to be included.*

We altered this sentence to offering an overstatement.

*3) Subsection “MBn are both pre- and post- synaptic to DAn forming axoaxonic reciprocal connections”, second paragraph: Please clarify why cAMP but not Ca^2+^ levels are elevated in MBn when DNa is stimulated.*

It has been reported before (Tomchik et al., 2009) and we observed again here, that bath applying dopamine does not generate calcium transients in KC. Why is this? This is likely due to the fact that dopamine receptors in KC trigger signaling pathways (including cAMP signaling) that do not alter ion channel function. An alternative but less likely explanation is that GCamP3.0 used in this and other studies (Tomchik et al., 2009) lacks the sensitivity to detect such changes in KC.

*4) It would be helpful to provide some general information about axonal-axonal reciprocal synapses in the Introduction, or at least to cite a reference, possibly a review article.*

We have included in the concluding remarks some general information and speculation about reciprocal connections and recurrent loops, and their roles in information processing and associative learning.